# 🏜️MoHAVE: Mixture of Hierarchical Audio-Visual Experts for Robust Speech Recognition

**Sungnyun Kim** [1]  **Kangwook Jang** [2]  **Sangmin Bae** [1]  **Sungwoo Cho** [1]  **Se-Young Yun** [1]

## Abstract

Audio-visual speech recognition (AVSR) has become critical for enhancing speech recognition in noisy environments by integrating both auditory and visual modalities. However, existing AVSR systems struggle to scale up without compromising computational efficiency. In this study, we introduce **MoHAVE (Mixture of Hierarchical Audio-Visual Experts)**, a novel robust AVSR framework designed to address these scalability constraints. By leveraging a Mixture-of-Experts (MoE) architecture, MoHAVE activates modality-specific expert groups, ensuring dynamic adaptation to various audio-visual inputs with minimal computational overhead. Key contributions of MoHAVE include: (1) a sparse MoE framework that efficiently scales AVSR model capacity, (2) a hierarchical gating mechanism that dynamically utilizes the expert groups based on input context, enhancing adaptability and robustness, and (3) remarkable performance across robust AVSR benchmarks, including LRS3 and MuAViC transcription and translation tasks, setting a new standard for scalable speech recognition systems.

## 1. Introduction

Audio-visual speech recognition (AVSR) (Noda et al., 2015; Afouras et al., 2018a; Ma et al., 2021b; Shi et al., 2022a; Hsu & Shi, 2022; Hu et al., 2023b) has emerged as a pivotal technology in enhancing the robustness and accuracy of speech recognition systems, particularly in noisy environments. By integrating auditory and visual modalities, AVSR leverages the complementary information from both speech signals and lip movements (Makino et al., 2019; Ren et al., 2021; Chen et al., 2023), offering significant advan-

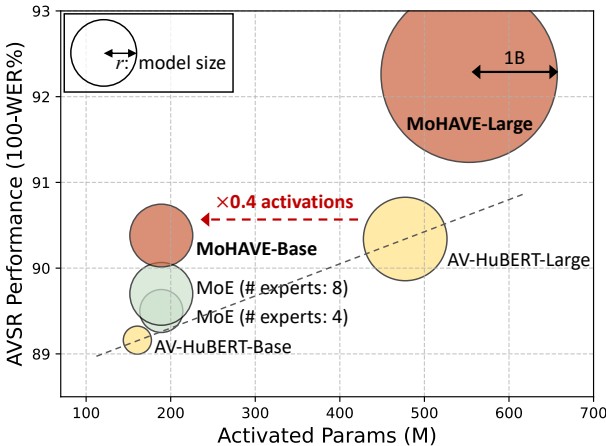

*Figure 1.* Comparison of AVSR models based on standard Transformers (AV-HuBERT, Shi et al. 2022a), MoE, and MoHAVE, evaluated under babble noise. The MoE structure boosts the model capacity while maintaining the number of activations. MoHAVE-BASE (359M) achieves similar performance to AV-HuBERT-LARGE (477M) while activating only 189M parameters.

tages over audio-only automatic speech recognition (ASR) approaches. This multimodal approach is indispensable in situations where auditory data alone is insufficient for reliable recognition.

Despite significant advances, AVSR systems have not kept pace with advancements in model scalability as seen in ASR (Radford et al., 2023) or large language models (LLMs) (Kaplan et al., 2020; Clark et al., 2022; Achiam et al., 2023). Contemporary AVSR models, such as AV-HuBERT (Shi et al., 2022a), AV-data2vec (Lian et al., 2023), and Auto-AVSR (Ma et al., 2023), generally employ fewer than 0.5B parameters, a stark contrast to large-scale ASR models like Whisper (Radford et al., 2023) or Seamless (Barrault et al., 2023) which boasts up to 1.6B and 2.3B parameters, respectively. This disparity is not merely a matter of size but reflects a fundamental challenge in AVSR scalability: increasing the model size often disproportionately enhances audio semantic understanding without similarly improving visual processing capabilities (Dai et al., 2024; Kim et al., 2024). Moreover, the computational complexity and latency of larger models pose challenges for efficient deployment, especially in scenarios where AVSR users often require

---

[1]KAIST AI, Republic of Korea [2]School of Electrical Engineering, KAIST, Republic of Korea. Correspondence to: Se-Young Yun <yunseyoung@kaist.ac.kr>.

*Proceedings of the 42$^{nd}$ International Conference on Machine Learning*, Vancouver, Canada. PMLR 267, 2025. Copyright 2025 by the author(s).

rapid processing and low latency. These factors make the integration of larger, more computationally intensive models impractical for many real-world applications.

To address the scalability challenges in AVSR systems, we leverage a sparse Mixture-of-Experts (MoE) architecture (Shazeer et al., 2017; Fedus et al., 2022) that activates only a subset of parameters (*i.e.,* experts) for efficient scaling of model capacity. Furthermore, recognizing the inherent bias in AVSR systems toward the audio modality, we find it essential to harness the full potential of both audio and video data. One approach is expert group specialization, also known as *hard routing* (Zhu et al., 2022; Li et al., 2023; Lee et al., 2025), which assigns specific roles to expert groups and manually activates them based on input types. While effective, this fixed activation strategy lacks adaptability, making it sub-optimal for AVSR where noise conditions and modality reliability vary. A more flexible routing mechanism is needed to dynamically utilize expert groups.

We thus propose a novel MoE framework, **MoHAVE**[2] (Mixture of Hierarchical Audio-Visual Experts), which employs a hierarchical gating mechanism with two-layer routers. MoHAVE introduces an inter-modal router that makes decision on utilizing audio and visual expert groups (§4.1). This dynamic routing adapts to input characteristics, specifically trained by our novel load biasing loss (§4.2). MoHAVE achieves state-of-the-art performance (§5.3) on the noisy LRS3 benchmark (Afouras et al., 2018b) and in multilingual tasks (Anwar et al., 2023). Empirical analysis shows that MoHAVE adaptively adjusts token distribution based on input context (§5.4), *e.g.,* visual expert group being more actively utilized under high auditory noise.

As shown in Figure 1, MoHAVE capitalizes on its increased model capacity to significantly enhance performance while maintaining efficiency. Unlike simple MoE implementations, which yield only modest gains over Transformers, our innovative expert group strategy unlocks substantial advancements in adaptability and robustness. Our main contributions are outlined as follows:

○ **MoE architecture for scaling AVSR systems**: We present MoHAVE, a framework that integrates a sparse MoE architecture to efficiently scale AVSR model capacity and optimally process audio and visual data.

○ **Hierarchical gating for adaptive expert utilization**: MoHAVE features a novel hierarchical gating mechanism that dynamically adjusts the usage of audio and visual expert groups based on input context, significantly improving adaptability and robustness.

○ **Robust AVSR performance**: Our model showcases substantial improvements across robust AVSR benchmarks including multilingual tasks, delivering high accuracy while maintaining computational overhead.

---

[2]MoHAVE is pronounced as *Mojave* Desert.

## 2. Related Work

### 2.1. Robustness of Audio-Visual Speech Recognition

The robustness of AVSR systems has significantly advanced by integrating auditory and visual cues to improve speech recognition, especially in noisy environments. Conventional ASR methods have evolved from relying solely on audio signals (Schneider et al., 2019; Gulati et al., 2020; Baevski et al., 2020; Hsu et al., 2021; Chen et al., 2022; Chiu et al., 2022; Radford et al., 2023) to incorporating visual data from speech videos (Makino et al., 2019). The multimodal AVSR methods (Pan et al., 2022; Shi et al., 2022a; Seo et al., 2023; Ma et al., 2023) have enhanced robustness under audio-corrupted conditions, leveraging visual details like speaker's face or lip movements as well as acoustic features of speech. These advancements have been driven by various approaches, including end-to-end learning frameworks (Dupont & Luettin, 2000; Ma et al., 2021b; Hong et al., 2022; Burchi & Timofte, 2023) and self-supervised pretraining (Ma et al., 2021a; Qu et al., 2022; Seo et al., 2023; Zhu et al., 2023; Kim et al., 2025), which focus on audio-visual alignment and the joint training of modalities (Zhang et al., 2023; Lian et al., 2023; Haliassos et al., 2023; 2024).

Furthermore, recent advancements in AVSR highlight the importance of visual understanding alongside audio (Dai et al., 2024; Kim et al., 2024). While initial research primarily targeted audio disturbances (Shi et al., 2022b; Hu et al., 2023c;d; Chen et al., 2023), latest studies increasingly focus on the visual robustness to address challenges such as real-world audio-visual corruptions (Hong et al., 2023; Wang et al., 2024; Kim et al., 2025) or modality asynchrony (Zhang et al., 2024; Fu et al., 2024; Li et al., 2024a). These efforts remark a shift towards a more balanced use of audio and visual modalities. Yet, there has been limited exploration in scaling model capacity or introducing innovative architectural designs, leaving room for further developments in AVSR system that can meticulously balance audio and visual modalities.

### 2.2. MoE for Language, Vision, and Speech Models

Mixture-of-Experts (MoE), first introduced by Jacobs et al. (1991), is a hybrid structure incorporating multiple sub-models, *i.e.,* experts, within a unified framework. The essence of sparsely-gated MoE (Shazeer et al., 2017; Lepikhin et al., 2021; Dai et al., 2022) lies in its routing mechanism where a learned router activates only a subset of experts for processing each token, significantly enhancing computational efficiency. Initially applied within LLMs using Transformer blocks, this structure has enabled unprecedented scalability (Fedus et al., 2022; Zoph et al., 2022; Jiang et al., 2024; Guo et al., 2025) and has been progressively adopted in multimodal models, especially in large vision-language models (LVLMs) (Mustafa et al., 2022; Lin

et al., 2024; McKinzie et al., 2025). Among these multimodal MoEs, Zhu et al. (2022); Shen et al. (2023); Li et al. (2023; 2024b) and Lee et al. (2025) share the similar philosophy to ours, assigning specific roles to each expert and decoupling them based on distinct modalities or tasks. These models design an expert to focus on specialized segments of input and enhance the targeted processing.

Beyond its applications in LLMs and LVLMs, the MoE framework has also been applied for speech processing (You et al., 2021; 2022; Hu et al., 2023a; Wang et al., 2023), where it has shown remarkable effectiveness in multilingual and code-switching ASR tasks. In addition, MoE has been employed in audio-visual models (Cheng et al., 2024; Wu et al., 2024), although they primarily focus on general video processing and not specifically on human speech videos. These approaches leverage MoE to model interactions between audio and visual tokens without directly processing multimodal tokens. Our research advances the application of the MoE framework to AVSR by designing a modality-aware hierarchical gating mechanism, which categorizes experts into audio and visual groups and effectively dispatches multimodal tokens to each expert group. This tailored design enhances the adaptability in managing audio-visual speech inputs, which often vary in complexity due to diverse noise conditions.

## 3. Preliminaries

### 3.1. Sparsely-gated MoE

In AVSR systems, the multimodal encoder processes a sequence of audio $\mathbf{a} = [a_1, a_2, \cdots]$ and video $\mathbf{v} = [v_1, v_2, \cdots]$ data into combined audio-visual embeddings $\text{Enc}(\mathbf{a}, \mathbf{v})$. These embeddings are utilized by the decoder to predict subsequent text tokens, where the predicted token is given by $text_{t+1} = \text{Dec}(\text{Enc}(\mathbf{a}, \mathbf{v}), text_t)$. Within the Transformer layer, $x_t$ is the intermediate representation of the token $text_t$, derived by cross-attending to the combined audio-visual embeddings from $\mathbf{a}$ and $\mathbf{v}$ (see Figure 2).

The integration of a sparsely-gated MoE framework (Shazeer et al., 2017; Lepikhin et al., 2021) leverages experts $\mathcal{E} = \{E_i\}$ to scale model capacity. Each token representation is routed to a selected subset of these experts through a learned gating mechanism. Specifically, the routing function $h(x) = W_r \cdot x$ assigns weights for each token, and the weight for expert $i$ is computed using a softmax function:

$$p_i(x) = \frac{\exp(h_i(x))}{\sum_{j=1}^{|\mathcal{E}|} \exp(h_j(x))}, \tag{1}$$

and the output $y$ is the aggregated result of computations from the top-$k$ selected experts:

$$y = \sum_{i \in \text{top}k(\mathcal{E})} \tilde{p}_i(x) E_i(x), \tag{2}$$

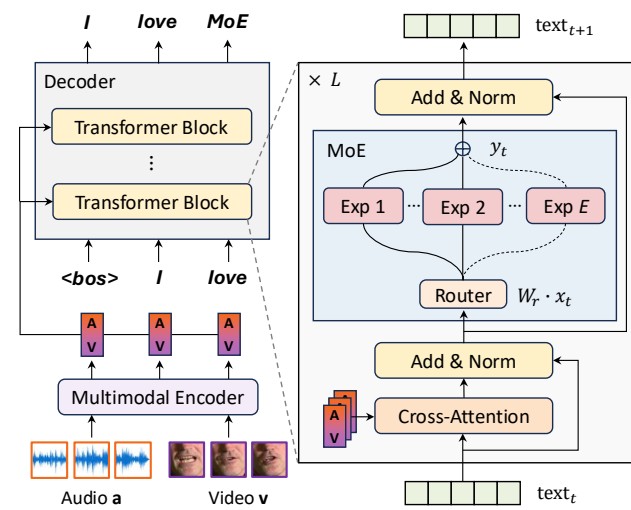

*Figure 2.* Overview of sparsely-gated MoE for AVSR. A select subset of experts are activated for each token representation ($x_t$).

where $\tilde{p}$ is the normalization of top-$k$ probabilities. Note that each expert follows the same structure as a feed-forward network (FFN) in a Transformer block. Figure 2 presents the overall MoE architecture and its token routing.

**Load balancing.** To mitigate the load imbalance issue commonly observed in the top-$k$ expert selection strategy, a load balancing loss has been implemented to encourage the balanced token load across all experts. Specifically, we use a differentiable load balancing loss (Fedus et al., 2022):

$$L_B = |\mathcal{E}| \cdot \sum_{i=1}^{|\mathcal{E}|} f_i \cdot P_i, \tag{3}$$

where $f_i$ denotes the frequency of expert $i$ being selected as top-1, averaged over all tokens within a batch $\mathcal{B}$,

$$f_i = \frac{1}{T} \sum_{x \in \mathcal{B}} \mathbb{1}\{\arg\max p(x) = i\} \tag{4}$$

and $P_i$ is the average assigned probability for expert $i$,

$$P_i = \frac{1}{T} \sum_{x \in \mathcal{B}} p_i(x) \tag{5}$$

with $T$ representing the total number of tokens.

An additional router z-loss (Zoph et al., 2022) is employed to stabilize the routing mechanism:

$$L_Z = \frac{1}{T} \sum_{x \in \mathcal{B}} \left( \log \sum_{i=1}^{|\mathcal{E}|} \exp(h_i(x)) \right)^2. \tag{6}$$

This sparse MoE structure ensures that token processing is efficiently managed across multiple experts, utilizing lower compute relative to its expansive capacity.

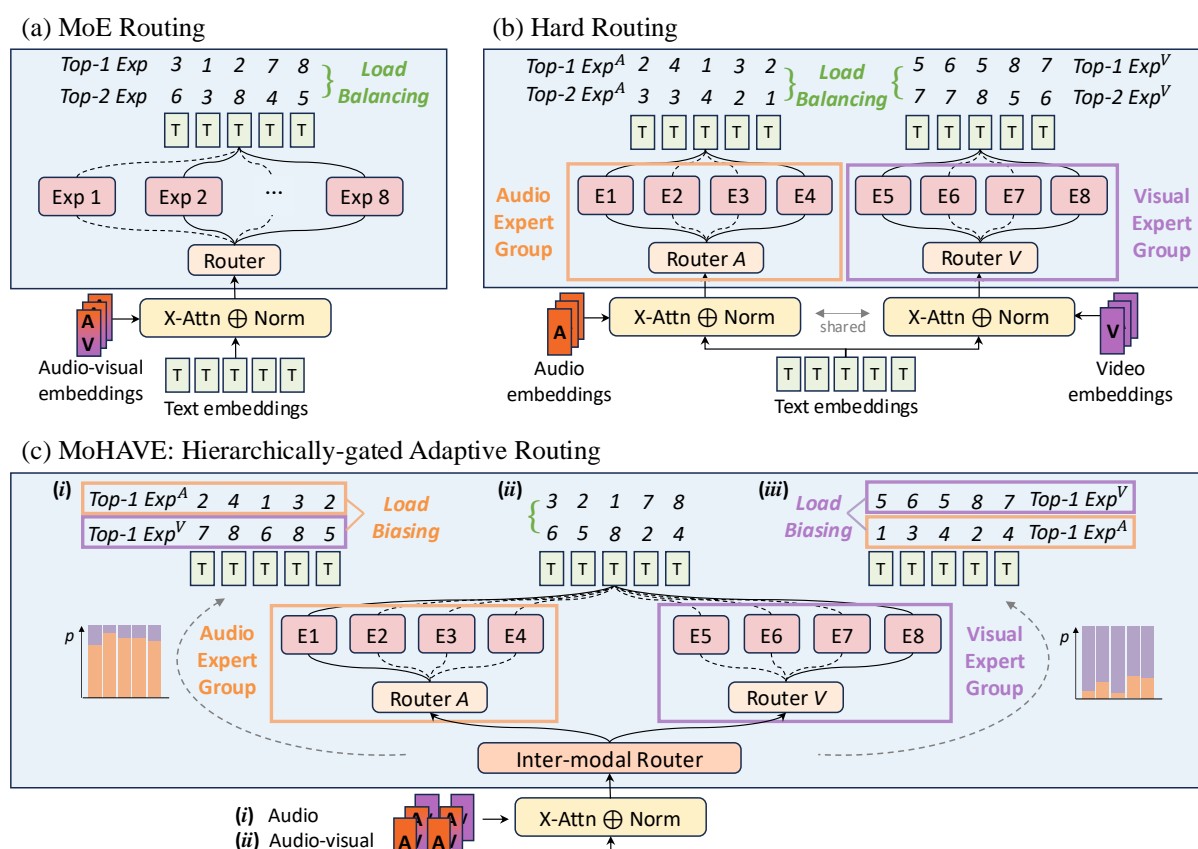

*Figure 3.* MoE-based routing strategies for AVSR. (a) A conventional MoE approach where a learned router selects the top-2 experts for each token, enforcing the balanced expert load. (b) Experts are explicitly divided into audio and visual groups, with manual activation based on the input modality. (c) MoHAVE introduces an inter-modal router that can dynamically assign weights to modality-specific expert groups, followed by intra-modal routers that select the top-1 expert within each group. The inter-modal router is trained by the load biasing loss that guides the expert group specialization.

## 3.2. Expert Group Specialization

To enhance expert management within the AVSR system, a *hard routing* technique can be used for expert group specialization. This approach is inspired by several practices in visual-language MoE models (Zhu et al., 2022; Li et al., 2023; Shen et al., 2023; Lee et al., 2025) where the role of experts is strictly defined by the input modality, eliminating the need for a trained router.

**Hard routing.** Our hard routing enforces modality-specific activation of expert groups: audio data activate only audio experts, and video data activate only visual experts. This segregation encourages the independent development of specialized expert groups. As suggested in V/T-MoE (Shen et al., 2023), once the group is activated, we use an intra-modal router for the modality-specific experts.

Figure 3(b) visualizes the hard routing mechanism with audio and visual expert groups. During training, audio or video sequence is randomly dropped, leading to subsets $\mathcal{A}$ and $\mathcal{V}$ within a batch $\mathcal{B}$, consisting of audio-only or video-only sequences, respectively. A token representation $x_t \in \mathcal{A}$ indicates that the cross-attention module processes the input $text_t$ with $\text{Enc}(\mathbf{a}, \mathbf{0})$—where the visual component is zeroed out—and vice versa for $x_t \in \mathcal{V}$. For these, we utilize two distinct intra-modal routing networks, $W_r^A$ and $W_r^V$:

$$h^A(x) = W_r^A \cdot x \quad \text{for } x \in \mathcal{A},$$
$$h^V(x) = W_r^V \cdot x \quad \text{for } x \in \mathcal{V}. \tag{7}$$

These routers calculate the weights $p^{\{A,V\}}(x)$ as in Eq. (1) within their respective expert group, either $\{E^A\}$ for audio or $\{E^V\}$ for visual. The output for each token is then

$$y = \begin{cases} \sum_{\text{top}k(E^A)} \tilde{p}_i^A(x) E_i^A(x) & \text{if } x \in \mathcal{A}, \\ \sum_{\text{top}k(E^V)} \tilde{p}_i^V(x) E_i^V(x) & \text{if } x \in \mathcal{V}. \end{cases} \tag{8}$$

For audio-visual sequences, outputs from both groups are averaged, with the top-$(k/2)$ experts from each group contributing to ensure balanced processing.

# 4. MoHAVE: Mixture of Hierarchical Audio-Visual Experts

Despite the benefits of hard routing in specializing expert groups according to decoupled input modality, it lacks the flexibility to autonomously determine the group utilization. In practice, the optimal balance between audio and visual groups varies depending on ambient conditions such as noise type and intensity (more detailed in Figure 4(b)). To address this limitation and enhance the model's adaptability, we introduce an adaptive routing mechanism with hierarchical gating (Jordan & Jacobs, 1994), providing a more dynamic approach to manage multimodal inputs.

Our hierarchical model, MoHAVE, features a two-layer routing mechanism: *inter-modal* and *intra-modal* routers, where the inter-modal router learns to assign appropriate weights to each modality-specific group. Figure 3(c) presents the overview of MoHAVE's routing strategy.

## 4.1. Hierarchical Gating Structure

The inter-modal router orchestrates the initial token distributions across expert groups. It generates logits through $u(x) = V_r \cdot x$ and determines the dispatch weights for group $i$ with $q_i(x) = \text{softmax}(u(x))_i$. This router dynamically selects the top-$m$ expert groups, and within those, the intra-modal routers select the top-$k$ experts, thus involving $m \times k$ experts in total. For practical efficiency, we set $k = 1$ for each group and modify the intra-modal router's probability distribution to a Kronecker delta, $\tilde{p}_{ij} \rightarrow \delta_{j,\text{argmax}(p_i)}$. The output from this layer integrates these selections:

$$y = \sum_{i \in \text{top}m(G)} \tilde{q}_i(x) \sum_{j \in \text{top}k(E_i)} \tilde{p}_{ij} E_{ij}(x) \qquad (9)$$

$$\rightarrow \sum_{i \in \text{top}m(G)} \tilde{q}_i(x) E_{ij}(x), \quad \text{where } j = \arg\max(p_i) \quad (10)$$

where $\tilde{q}$ is the normalization of $q$ across top-$m$ probabilities, $G$ is the number of expert groups, and $E_{ij}(x)$ denotes the output from the $j$-th expert in the $i$-th group.

Focusing on audio-visual applications, we designate two expert groups: audio and visual. Each token $x$, regardless of its modality, is processed by the intra-modal routing networks of both groups, *i.e.*, $[h^A(x), h^V(x)] = [W_r^A, W_r^V] \cdot x$. The frequencies $f^{\{A,V\}}$ and probabilities $P^{\{A,V\}}$ for selecting experts are computed in the same manner as Eq. (4)–(5) for all $x \in \mathcal{B}$. Thus, the load balancing loss can be computed for both groups:

$$L_B = |E^A| \cdot \sum_{j=1}^{|E^A|} f_j^A \cdot P_j^A + |E^V| \cdot \sum_{j=1}^{|E^V|} f_j^V \cdot P_j^V \quad (11)$$

where $f_j^A$ and $f_j^V$ denote the frequencies of token assignments to audio and visual experts, respectively.

Table 1. Computational cost of AV-HuBERT and MoHAVE in FLOPs, along with their sizes (activated and total parameters).

| Model | Params Act. & Total | Compute (GFLOPs / seq) | Compute / FFN (MFLOPs / seq) |
|---|---|---|---|
| AV-HuBERT-BASE | 161M & 161M | 12.1 | 472 |
| MoHAVE-BASE | 189M & 359M | 14.8 | 921 |
| AV-HuBERT-LARGE | 477M & 477M | 32.2 | 839 |
| MoHAVE-LARGE | 553M & 1.0B | 39.3 | 1,630 |

## 4.2. Group-level Load Biasing Loss

To autonomously manage the expert group loads without manual (de-)activation as hard routing, we introduce a biasing loss that directs the load towards a certain group. This load biasing loss encourages the inter-modal router to assign higher weights to $E^A$ experts for audio sequences and to $E^V$ experts for video sequences. For audio sequences within a sub-batch $\mathcal{A}$, the frequency and average probability of selecting the $i$-th group is calculated as follows:

$$g_i^A = \frac{1}{|\mathcal{A}|} \sum_{x \in \mathcal{A}} \mathbb{1}\{\arg\max q(x) = i\}, \qquad (12)$$

$$Q_i^A = \frac{1}{|\mathcal{A}|} \sum_{x \in \mathcal{A}} q_i(x). \qquad (13)$$

Similar calculations for $g_i^V$ and $Q_i^V$ are made for video sequences $x \in \mathcal{V}$. We designate the first group as audio experts and the second group as video experts, then the load biasing loss is defined as:

$$L_S = L_S^A + L_S^V = (1 - g_1^A \cdot Q_1^A) + (1 - g_2^V \cdot Q_2^V). \ (14)$$

Note that $L_S^A$ and $L_S^V$ are only computed over $x \in \mathcal{A}$ and $x \in \mathcal{V}$, respectively.

For sequences containing both audio and video, we exclude them from the load biasing loss calculation but incorporate them into the load balancing. Although these tokens are uniformly dispatched on average, the inter-modal router finds the optimal strategy for each token based on its characteristics. Empirically, we find that MoHAVE learns to assign greater weight to the visual expert group for audio-visual inputs under high auditory noise, and to the audio expert group for less noisy inputs (see §5.4 for details), demonstrating the model's adaptability under various noisy conditions.

The overall loss function, combining the cross-entropy (CE) for token prediction, is formulated as:

$$L_{tot} = L_{CE} + c_B L_B + c_S L_S + c_Z L_Z. \qquad (15)$$

Here, $c_B$ and $c_Z$ are set to 1e-2 and 1e-3, respectively, in line with (Fedus et al., 2022; Zoph et al., 2022), and $c_S$ is also set at 1e-2.

# 5. Experiments and Results

## 5.1. Implementation Details

**Datasets.** For the robust AVSR benchmark, we utilize the LRS3 dataset (Afouras et al., 2018b), which consists of 433 hours of audio-visual speech from 5,000+ speakers. Following the experimental setup of Shi et al. (2022b), we extract audio noise samples from the MUSAN (Snyder et al., 2015) dataset, targeting different noise types such as *babble*, *music*, and *natural* noises, along with *speech* noise from LRS3. These noises are randomly augmented into the audio data, corrupting 25% of the training set with a signal-to-noise ratio (SNR) sampled from $\mathcal{N}(0, 5)$. We measure performance using the word error rate (WER), primarily under noisy conditions with SNRs of $\{-10, -5, 0, 5, 10\}$ dB, specifically N-WER (Kim et al., 2024) which highlights the significance of visual cues in noise-corrupted environments.

For multilingual evaluations, the MuAViC dataset (Anwar et al., 2023) is used, featuring 1,200 hours of audio-visual content from 8,000+ speakers across 9 languages, sourced from LRS3-TED (Afouras et al., 2018b) and mT-EDx (Elizabeth et al., 2021). We use 8 languages (excluding English) for multilingual AVSR and 6 languages for X-to-English audio-visual speech-to-text translation (AVS2TT) tasks. We assess the models using WER for transcription and the BLEU score (Papineni et al., 2002) for translation.

**MoHAVE model description.** Our MoHAVE framework is developed in two configurations: BASE and LARGE. The BASE model consists of 12 Transformer (Vaswani et al., 2017) encoder layers and 6 decoder layers, while the LARGE model incorporates 24 encoder layers and 9 decoder layers. Both models' audio-visual encoders are derived from the AV-HuBERT-BASE/-LARGE models, pretrained on a noise-augmented corpus of LRS3 (Afouras et al., 2018b) + VoxCeleb2 (Chung et al., 2018). Our MoE implementation activates top-2 out of 8 experts in every FFN layer within the decoder (Jiang et al., 2024), while the hierarchical architecture engages the top-1 expert from each audio and visual group. To facilitate the expert group specialization, load biasing is used with audio or video randomly dropped in 25% probability.

As summarized in Table 1, the BASE model of MoHAVE holds 359M parameters, and the LARGE configuration contains 1B. Specifically, for MoHAVE-BASE, the encoder accounts for 103M parameters, and the decoder 256M, whereas in LARGE, the encoder holds 325M, and the decoder 681M. Despite its larger model capacity, due to the sparse activation of these parameters, only about half are active during token processing, amounting to 189M for BASE and 553M for LARGE model. This setup ensures computational efficiency which is comparable to the smaller AV-HuBERT counterparts. For more details on the model description, refer to Appendix A.1.

## 5.2. Computation Cost

Table 1 summarizes the parameter sizes and computational costs of MoHAVE. To assess actual computation costs when processing inputs, we measure floating point operations per second (FLOPs) using an audio-visual sequence of 500 frames with 50 text tokens. The entire compute cost for AV-HuBERT-BASE and MoHAVE-BASE are 12.1 GFLOPs and 14.8 GFLOPs, respectively, while for LARGE, the computes are 32.2 GFLOPs and 39.3 GFLOPs. This indicates a slight increase in FLOPs for MoHAVE, primarily due to the MoE layers replacing FFNs in the decoder. Although the MoE layers require roughly twice the computation cost of standard FFNs (refer to Compute / FFN), the encoder and attention layers in the decoder remain unchanged. Consequently, the overall computational cost remains comparable to AV-HuBERT counterparts, ensuring scalability without significant computation overhead.

## 5.3. Robust AVSR Benchmark Results

Table 2 presents MoHAVE's robust performance on the AVSR benchmark under diverse noisy conditions, demonstrating exceptional robustness across different noise types and SNR levels: **N-WER of 5.8% for BASE and 4.5% for LARGE**. This substantiates the model's potential for effectively scaling AVSR systems without incurring significant computational costs. The results also reveal that simple MoE implementations (AV-MoE in Table 2), despite their larger capacity, fail to achieve remarkable gains. Instead, the key improvement stems from leveraging expert group specialization, as evidenced by the effectiveness of hard routing. By splitting experts into audio and visual groups, MoE is enabled with more targeted and effective processing of multimodal inputs, leading to substantial performance enhancements. Without our load biasing loss, MoHAVE loses its group specialization capability, comparable to the performance of simple AV-MoEs.

Building upon this expert group strategy, MoHAVE enhances its adaptability through dynamically determining the usage of each group. This adaptive routing approach allows the model to flexibly adjust to varying audio-visual scenarios, contributing to consistent gains in robustness across the benchmark, as detailed in Table 2. An in-depth analysis of this hierarchical gating approach and its impact on token dispatching is discussed in §5.4, underscoring its critical role in advancing MoHAVE's capabilities in various AVSR environments.

**Comparison with state-of-the-art AVSR methods.** Table 3 shows how our MoHAVE decoder, when integrated with a range of audio-visual encoders, consistently improves

*Table 2.* Audio-visual speech recognition performance (WER %) on the LRS3 dataset (Afouras et al., 2018b). The number of parameters for each model includes both encoder and decoder. For evaluation, augmented noise is sampled from the MUSAN dataset (Snyder et al., 2015), while speech noise is sampled from the held-out set of LRS3. N-WER (Kim et al., 2024) averages the results across all four noise types and five signal-to-noise ratios, and C-WER indicates the result with a clean audio signal.

| Model | # Experts | Specialized Groups | Activated Params | Total Params | SNR = {−10, −5, 0, 5, 10} | | | | N-WER | C-WER |
|---|---|---|---|---|---|---|---|---|---|---|
| | | | | | babble | speech | music | natural | | |
| AV-HuBERT-BASE | - | - | 161M | 161M | 10.8 | 4.9 | 5.6 | 5.1 | 6.6 | 2.1 |
| AV-MoE-BASE | 4 | ✗ | 189M | 246M | 10.5 | 4.5 | 5.3 | 5.0 | 6.3 | 2.0 |
| AV-MoE-BASE | 8 | ✗ | 189M | 359M | 10.5 | 4.5 | 5.3 | 4.9 | 6.3 | 2.1 |
| (+) Hard Routing | 8 | ✓ | 189M | 359M | 9.9 | 4.4 | 5.0 | 4.6 | 5.9 | 2.0 |
| MoHAVE-BASE (ours) | 2×4 (H) | ✓ | 189M | 359M | **9.6** | **4.2** | **4.7** | **4.5** | **5.8** | **1.8** |
| (−) Load Biasing | 2×4 (H) | ✗ | 189M | 359M | 10.3 | 4.4 | 5.2 | 4.9 | 6.2 | 2.0 |
| AV-HuBERT-LARGE | - | - | 477M | 477M | 9.7 | 4.6 | 4.4 | 4.1 | 5.7 | 1.4 |
| AV-MoE-LARGE | 4 | ✗ | 553M | 704M | 10.1 | 3.8 | 4.6 | 4.3 | 5.7 | 1.8 |
| AV-MoE-LARGE | 8 | ✗ | 553M | 1.0B | 10.1 | 3.8 | 4.6 | 4.2 | 5.7 | 1.8 |
| (+) Hard Routing | 8 | ✓ | 553M | 1.0B | 8.3 | 3.3 | 4.0 | 3.7 | 4.8 | 1.5 |
| MoHAVE-LARGE (ours) | 2×4 (H) | ✓ | 553M | 1.0B | **7.7** | **3.0** | **3.7** | **3.4** | **4.5** | 1.5 |
| (−) Load Biasing | 2×4 (H) | ✗ | 553M | 1.0B | 9.9 | 3.6 | 4.6 | 4.3 | 5.6 | 1.7 |

performance compared to existing state-of-the-art methods. While BRAVEn (Haliassos et al., 2024) typically struggles in noisy multimodal scenarios—due to its original design focused on handling unimodal tasks—MoHAVE boosts its accuracy.

Other recent approaches have advanced by utilizing the noise-augmented AVSR encoder (Shi et al., 2022a), such as additionally learning temporal dynamics with cross-modal attention modules (CMA) (Kim et al., 2024). MIR-GAN (Hu et al., 2023b), UniVPM (Hu et al., 2023c), and CMA are all built upon AV-HuBERT-LARGE, matching the architecture and activated parameter count with the dense (non-MoE) baseline in Table 2. When paired with an AV-HuBERT encoder and trained through the CMA's self-supervised learning, MoHAVE achieves a remarkable performance: **N-WER of 4.2%**.

## 5.4. Expert and Group Load Analysis

**MoHAVE's expert load distribution.** Figure 4(a) illustrates the expert load distribution of MoHAVE according to input types: audio-visual, audio-only, and video-only sequences. For audio-visual inputs, all experts from both the audio and visual groups are selected at similar frequencies, with some layer-dependent variations. In contrast, when processing audio-only sequences, the model predominantly activates the audio expert group, while for video-only sequences, the visual expert group is mainly utilized. This distribution validates the effectiveness of our load biasing loss in guiding the inter-modal router to assign appropriate weights based on input modality.

**Expert group utilization in noisy AVSR.** To analyze the effectiveness of hierarchical gating in AVSR, we first examine the limitations of hard routing (§3.2) under noisy condi-

*Table 3.* Performance comparison on the noisy LRS3 benchmark with prior works. We present average N-WER, with babble (B), speech (S), music (M), and natural (N) noise types. BRAVEn is implemented with separate ASR and VSR encoders, combined and jointly trained with a decoder.

| Method | B | S | M + N | Avg |
|---|---|---|---|---|
| *Joint ASR and VSR encoders* | | | | |
| BRAVEn (Haliassos et al., 2024) | 13.5 | 15.7 | 7.4 | 11.0 |
| + MoHAVE (**ours**) | 12.3 | 13.4 | 6.8 | **9.8** |
| *Noise-augmented AVSR encoder* | | | | |
| AV-HuBERT (Shi et al., 2022b) | 9.7 | 4.6 | 4.2 | 5.7 |
| + MIR-GAN (Hu et al., 2023b) | - | - | - | 5.6 |
| + UniVPM (Hu et al., 2023c) | 9.3 | 4.1 | 3.6 | 5.2 |
| + CMA (Kim et al., 2024) | 8.1 | 2.9 | 3.7 | 4.6 |
| + MoHAVE (**ours**) | 7.7 | 3.0 | 3.6 | **4.5** |
| + CMA + MoHAVE (**ours**) | 7.3 | 2.8 | 3.3 | **4.2** |

tions. Since hard routing relies on manually (de-)activating the audio and visual groups, for audio-visual inputs, we assign a fixed equal weight ($p^A, p^V = 0.5$) to both groups. However, this equal weighting may not always be optimal in varying environments, such as noise type or intensity.

As shown in Figure 4(b), increasing reliance on the audio group under babble noise degrades performance, with an optimal weight for the audio group being $0.3$. Unlike babble noise, which confuses the model with multiple overlapping speech signals, natural noise is more distinct from speech, leading to a higher reliance on the audio group ($p^A \geq 0.5$) preferable. These results indicate that an ideal routing strategy for audio-visual data should be dynamically adjusted.

Figure 5 further illustrates MoHAVE's group load distribution across different noise levels. The model adaptively adjusts its reliance between the audio and visual expert groups—under high noise conditions (low SNRs), it shifts

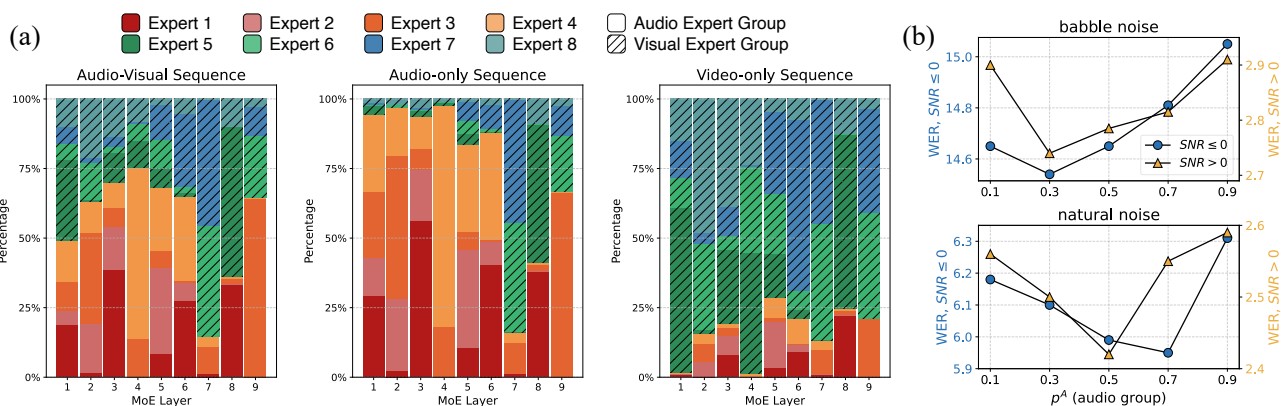

*Figure 4.* (a) Expert load distribution in MoHAVE according to input modalities, with expert selection frequencies weighted by the inter-modal router's output probability. (b) Performance of the hard routing strategy under different weight assignments to audio expert group. The visual expert group is weighted by $p^V = 1 - p^A$.

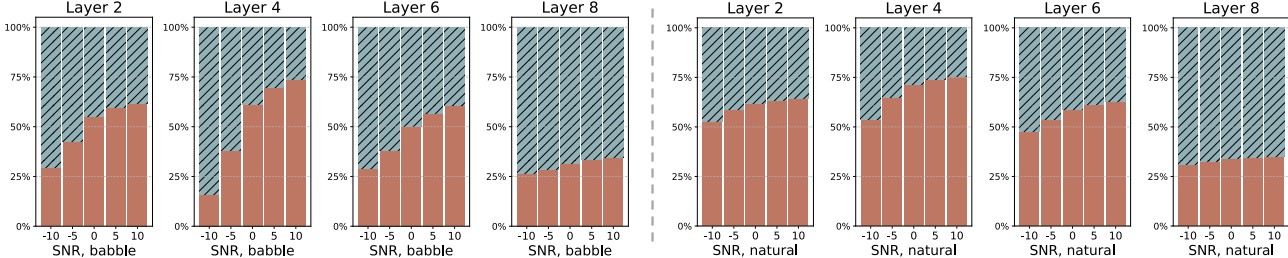

*Figure 5.* Expert load distribution in MoHAVE for the audio group (solid bars) and visual group (dashed bars) across noisy audio-visual sequences under babble (left) and natural (right) noise. Full layer-wise results are provided in Appendix C.1.

more tokens to the visual group, while in cleaner conditions (high SNRs), the audio group is more actively utilized. This behavior also adjusts to noise types, as observed with babble and natural noise, demonstrating the MoHAVE's adaptability and robustness.

### 5.5. Multilingual Audio-Visual Speech Tasks

MoEs have demonstrated effectiveness in multilingual speech tasks (Hu et al., 2023a; Wang et al., 2023), as MoE is capable of enabling more diverse routing paths for different language tokens. To evaluate MoHAVE's multilingual capabilities, we train a multilingual model and assess its performance on the MuAViC benchmark (Anwar et al., 2023), evaluating separately for each language. Following Han et al. (2024), we introduce multilingual babble noise at SNR 0 dB to 50% of the input samples during training, where the noise clips are sampled from the MuAViC train set. For inference, we apply beam search with a beam size of 5 and normalize text by punctuation removal and lower-casing before calculating WER. For AVS2TT evaluation, we use SacreBLEU (Post, 2018) with its built-in *13a* tokenizer. To simulate noisy test conditions, we inject babble noise sampled from the MuAViC test set.

Table 4 summarizes the results, where MoHAVE-LARGE

achieves superior performance in both AVSR and AVS2TT. Whisper (Radford et al., 2023), a leading multilingual ASR model, is known to perform poorly in noisy setup due to its lack of visual understanding for robustness. While multilingual AV-HuBERT (Anwar et al., 2023) underperforms the state-of-the-art models like XLAVS-R (Han et al., 2024), we have re-implemented it using the pretrained model from Choi et al. (2024), which has been pretrained on a significantly larger dataset including 7,000 hours of speech in 100+ languages. This model outperforms (38.9% average WER) or remains competitive (18.5% average BLEU) with much larger XLAVS-R 2B. When integrated with this version, MoHAVE further improves performance, achieving **37.4% average WER** and **19.5% average BLEU**, setting new benchmarks in almost every language being evaluated.

## 6. Conclusion

In this study, we propose MoHAVE, a hierarchical MoE framework for AVSR, designed to enhance scalability and robustness. By training an inter-modal router that dynamically assigns weights to audio and visual expert groups, MoHAVE enables an adaptive group selection based on input context. Evaluations on robust AVSR benchmarks demonstrate its state-of-the-art performance, with superior noise

*Table 4.* Multilingual audio-visual speech task performance, with non-English speech recognition (WER) and X-En speech-to-text translation (BLEU score), in a noisy environment with multilingual babble noise (SNR = 0). [†]Results obtained from Han et al. (2024). [‡]Re-implemented using the pretrained model from Choi et al. (2024).

| Model | Pretrain data | Source | | | | | | | | Avg |
|---|---|---|---|---|---|---|---|---|---|---|
| | | Ar | De | El | Es | Fr | It | Pt | Ru | |
| *Speech Recognition, Test WER ↓* | | | | | | | | | | |
| Whisper large-v2 (Radford et al., 2023) | *680k hrs, 100+ langs* | 197.9 | 244.4 | 113.3 | 116.3 | 172.3 | 172.4 | 223.6 | 126.2 | 170.8 |
| u-HuBERT[†] (Hsu & Shi, 2022) | *1.7k hrs, English* | 102.3 | 73.2 | 69.7 | 43.7 | 43.2 | 48.5 | 47.6 | 67.0 | 61.9 |
| mAV-HuBERT (Anwar et al., 2023) | *1.7k hrs, English* | **82.2** | 66.9 | 62.2 | 40.7 | 39.0 | 44.3 | 43.1 | 43.1 | 52.7 |
| XLS-R 300M[†] (Babu et al., 2022) | *1.2k hrs, 9 langs* | 97.3 | 69.8 | 74.8 | 47.6 | 37.1 | 47.9 | 54.4 | 59.8 | 61.1 |
| XLAVS-R 300M (Han et al., 2024) | *8.3k hrs, 100+ langs* | 91.9 | 53.5 | 49.6 | 28.8 | 29.3 | 32.2 | 32.5 | 46.1 | 45.5 |
| XLAVS-R 2B (Han et al., 2024) | *1.2k hrs, 9 langs* | 93.5 | 58.5 | 38.6 | 23.9 | 23.5 | 24.6 | 26.1 | 41.0 | 41.2 |
| mAV-HuBERT[‡] | *7.0k hrs, 100+ langs* | 88.7 | 51.3 | 37.2 | 20.7 | 22.6 | 24.2 | 23.8 | 42.4 | 38.9 |
| mAV-HuBERT + Hard Routing | *7.0k hrs, 100+ langs* | 93.4 | 49.3 | 35.7 | 20.3 | 23.6 | 23.4 | 24.1 | 44.7 | 39.3 |
| MoHAVE-LARGE (**ours**) | *7.0k hrs, 100+ langs* | 92.9 | **47.3** | **35.3** | **18.7** | **21.2** | **21.6** | **21.9** | 40.6 | **37.4** |
| *X-En Speech-to-Text Translation, Test BLEU ↑* | | | | | | | | | | |
| Whisper large-v2 (Radford et al., 2023) | *680k hrs, 100+ langs* | - | - | 0.1 | 0.4 | 0.7 | 0.1 | 0.1 | 0.2 | 0.3 |
| mAV-HuBERT (Anwar et al., 2023) | *1.7k hrs, English* | - | - | 4.2 | 12.8 | 15.0 | 12.5 | 14.8 | 4.6 | 10.7 |
| XLAVS-R 300M (Han et al., 2024) | *8.3k hrs, 100+ langs* | - | - | 13.2 | 17.4 | 23.8 | 18.7 | 21.8 | 9.4 | 17.4 |
| XLAVS-R 2B (Han et al., 2024) | *1.2k hrs, 9 langs* | - | - | **15.7** | 19.2 | 24.6 | 20.1 | 22.3 | **10.4** | 18.7 |
| mAV-HuBERT[‡] | *7.0k hrs, 100+ langs* | - | - | 8.9 | 21.5 | 26.5 | 21.2 | 24.2 | 8.8 | 18.5 |
| mAV-HuBERT + Hard Routing | *7.0k hrs, 100+ langs* | - | - | 6.7 | 19.9 | 24.7 | 19.6 | 23.0 | 7.2 | 16.8 |
| MoHAVE-LARGE (**ours**) | *7.0k hrs, 100+ langs* | - | - | 11.4 | **22.3** | **27.1** | **22.1** | **25.1** | 9.2 | **19.5** |

resilience, further supported by flexible expert load distributions across diverse noisy conditions. This work establishes an adaptive modality-aware MoE paradigm, advancing larger-scale multimodal speech recognition systems.

## Acknowledgements

This work was supported by Institute of Information & communications Technology Planning & Evaluation (IITP) grant funded by the Korea government (MSIT) [No.2022-0-00641, XVoice: Multi-Modal Voice Meta Learning, 90%] and [No.2019-0-00075, Artificial Intelligence Graduate School Program (KAIST), 10%].

## Impact Statement

MoHAVE addresses the scalability and robustness challenges in multimodal speech processing, enabling more efficient and adaptive expert selection. This advancement highlights the potential for further scaling AVSR models in terms of capacity, training, and datasets while extending to more diverse real-world applications without incurring excessive computational costs.

While our research contributes to the broader field of multimodal learning and robust speech recognition, we do not foresee significant ethical concerns beyond those inherent to AVSR systems, such as potential linguistic biases in speech models trained on imbalanced datasets or privacy issues when handling human speech video data. Responsible data use and fairness in model deployment remain critical in AVSR systems for ensuring equitable applications.

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

# ⛰️ MoHAVE: Mixture of Hierarchical Audio-Visual Experts for Robust Speech Recognition

## Supplementary Material

## A. Experimental Setup

### A.1. Model Description

As described in §5.1, MoHAVE is implemented in two configurations: BASE and LARGE, following the architecture of AV-HuBERT-BASE and AV-HuBERT-LARGE, respectively. The encoder maintains the same structure as AV-HuBERT, while the decoder incorporates MoE layers by replacing every feed-forward network (FFN) layer with expert modules. Each expert in the MoE layers follows the same bottleneck structure with FFN, consisting of two fully-connected layers with an activation function.

To encourage expert group specialization, we apply load biasing, where either audio or video is randomly dropped with a probability of 25%. This allows the model to learn modality-aware expert utilization. For expert selection, a router network assigns tokens to a subset of experts, ensuring efficient computation. The router probabilities are always normalized as sum to 1 when computing the output $y$. The routing networks $V_r$ and $W_r$ are parameterized as matrices, with dimensions matching the hidden dimension size by the number of experts.

For comparison, we evaluate multiple MoE-based AVSR models:

○ AV-MoE: A simple MoE implementation over AV-HuBERT, activating top-2 out of 4 or 8 experts per token. We follow the same implementation of sparse MoE as (Dai et al., 2022; Jiang et al., 2024).

○ AV-MoE with Hard Routing: Uses top-2 out of 4 experts for unimodal inputs (audio-only or video-only). For multimodal (audio-visual) inputs, it activates top-1 from each expert group and averages their outputs. This model does not have an explicit router for groups, but within each group, there is an intra-modal router, *i.e.*, $W_r^A$ or $W_r^V$.

○ MoHAVE: Employs top-1 expert per group, with an inter-modal router dynamically adjusting group weight assignments and an intra-modal router uniformly dispatching the tokens to modality-specific experts.

### A.2. LRS3 Benchmark Experiments

We initialize our model using the pretrained checkpoint from (Shi et al., 2022a) and fine-tune it on the LRS3 train set for 120K steps. The encoder remains frozen for the first 90K steps, allowing only the AVSR decoder to be trained, after which the entire model is fine-tuned for the remaining 30K steps. Our fine-tuning setup follows the configurations from (Shi et al., 2022b). We employ a sequence-to-sequence negative log-likelihood loss for predicting the next text token, without using connectionist temporal classification (CTC) decoding (Watanabe et al., 2017). The Adam optimizer (Kingma, 2014) is used with a learning rate of 5e-4 and a polynomial decay schedule with an initial warmup. Each training step processes 8,000 audio-visual frames, equivalent to 320 seconds of speech data.

For inference, we use beam search with a beam size of 50. The AVSR performance is evaluated using word error rate (WER) across five signal-to-noise ratio (SNR) levels: $\{-10, -5, 0, 5, 10\}$ (lower value means higher noise level). We use audio noise sampled from MUSAN (babble, music, natural) and LRS3 speech noise, ensuring no speaker overlap between training and test sets. Since Table 2 presents SNR-averaged results for each noise type, we provide the full results across all SNR levels in Table 5. MoHAVE-LARGE achieves 5.0% WER on LRS3 with speech noise at SNR=$-10$, yielding a 56.1% relative WER improvement over AV-HuBERT-LARGE, 36.7% over AV-MoE-LARGE, and 25.4% over the hard-routing variant. This indicates that MoHAVE correctly predicts over half of the words that AV-HuBERT misses.

### A.3. MuAViC Benchmark Experiments

We evaluate MoHAVE on the MuAViC benchmark (Anwar et al., 2023) for multilingual AVSR and X-to-English AVS2TT tasks. For multilingual AVSR, the dataset includes 8 non-English languages: Arabic (Ar), German (De), Greek (El), Spanish

*Table 5.* Audio-visual speech recognition performance (WER %) on the LRS3 dataset (Afouras et al., 2018b). The number of parameters for each model includes both encoder and decoder. For evaluation, augmented noise is sampled from the MUSAN dataset (Snyder et al., 2015), while speech noise is sampled from the held-out set of LRS3. AV-MoE and MoHAVE use 8 experts.

| Model | Babble, SNR (dB) = | | | | | | Speech, SNR (dB) = | | | | | | Music, SNR (dB) = | | | | | | Natural, SNR (dB) = | | | | | |
|---|---|---|---|---|---|---|---|---|---|---|---|---|---|---|---|---|---|---|---|---|---|---|---|---|
| | -10 | -5 | 0 | 5 | 10 | **AVG** | -10 | -5 | 0 | 5 | 10 | **AVG** | -10 | -5 | 0 | 5 | 10 | **AVG** | -10 | -5 | 0 | 5 | 10 | **AVG** |
| AV-HuBERT-Base | 27.6 | 14.1 | 6.1 | 3.8 | 2.7 | 10.8 | 8.6 | 5.8 | 3.9 | 3.3 | 2.8 | 4.9 | 12.2 | 6.4 | 3.8 | 2.8 | 2.5 | 5.6 | 10.9 | 5.4 | 4.0 | 2.8 | 2.3 | 5.1 |
| AV-MoE-Base | 26.3 | 13.7 | 6.2 | 3.5 | 2.7 | 10.5 | 8.4 | 5.3 | 3.4 | 2.8 | 2.3 | 4.5 | 11.5 | 6.1 | 3.7 | 2.7 | 2.5 | 5.3 | 9.7 | 6.0 | 3.4 | 2.8 | 2.5 | 4.9 |
| (+) Hard Routing | 25.2 | 13.2 | 5.6 | 3.2 | 2.4 | 9.9 | 8.2 | 5.2 | 3.4 | 2.7 | 2.3 | 4.4 | 11.0 | 5.3 | 3.6 | 2.6 | 2.2 | 5.0 | 9.4 | 5.3 | 3.2 | 2.5 | 2.3 | 4.6 |
| MoHAVE-Base | 25.3 | **12.2** | **5.3** | **2.9** | **2.3** | **9.6** | **7.9** | **5.1** | **3.3** | **2.4** | 2.3 | **4.2** | **10.3** | 5.6 | 3.3 | **2.3** | **2.0** | **4.7** | 9.7 | **5.1** | 3.2 | **2.4** | **2.2** | **4.5** |
| (−) Load Biasing | 26.5 | 13.6 | 5.6 | 3.2 | 2.4 | 10.3 | 8.2 | 5.2 | 3.5 | 2.6 | 2.4 | 4.4 | 11.1 | 6.4 | 3.4 | 2.7 | 2.6 | 5.2 | 10.3 | 5.6 | 3.4 | 2.7 | 2.4 | 4.9 |
| AV-HuBERT-Large | 27.0 | 12.4 | 4.7 | 2.4 | 1.8 | 9.7 | 11.4 | 4.6 | 2.9 | 2.2 | 1.8 | 4.6 | 10.5 | 4.9 | 2.9 | 2.0 | 1.6 | 4.4 | 9.6 | 4.7 | 2.5 | 2.0 | 1.8 | 4.1 |
| AV-MoE-Large | 28.1 | 12.5 | 5.0 | 2.7 | 2.1 | 10.1 | 7.9 | 4.0 | 2.9 | 2.4 | 2.0 | 3.8 | 10.4 | 5.4 | 2.9 | 2.3 | 1.9 | 4.6 | 8.9 | 4.8 | 3.1 | 2.0 | 2.0 | 4.2 |
| (+) Hard Routing | 22.9 | 10.8 | 3.8 | 2.4 | 1.8 | 8.3 | 6.7 | 3.9 | 2.4 | 1.8 | 1.8 | 3.3 | 9.9 | 4.3 | 2.3 | 1.8 | 1.9 | 4.0 | 8.3 | 4.1 | 2.4 | 1.9 | 1.7 | 3.7 |
| MoHAVE-Large | **21.0** | **9.8** | 4.1 | **2.2** | **1.6** | **7.7** | **5.0** | **3.6** | **2.3** | 2.0 | 1.9 | **3.0** | **8.2** | **4.0** | 2.6 | **1.8** | **1.8** | **3.7** | **7.3** | **3.7** | 2.6 | **1.9** | **1.6** | **3.4** |
| (−) Load Biasing | 27.8 | 12.4 | 4.5 | 2.6 | 2.0 | 9.9 | 6.7 | 4.0 | 3.1 | 2.1 | 1.9 | 3.6 | 10.6 | 5.3 | 3.0 | 2.1 | 1.8 | 4.6 | 9.9 | 4.9 | 2.8 | 2.1 | 2.0 | 4.3 |

(Es), French (Fr), Italian (It), Portuguese (Pt), and Russian (Ru), encompassing approximately 700 hours of training data from 3,700 speakers. For X-En AVS2TT, the dataset covers 6 languages: Greek, Spanish, French, Italian, Portuguese, and Russian, where each sample includes audio-visual speech with corresponding English transcriptions.

A single multilingual model is trained for each task, capable of detecting the source language and generating target transcriptions accordingly. The evaluation is conducted on each language separately, as seen in Table 4. Using the pretrained multilingual AV-HuBERT from (Choi et al., 2024), we fine-tune the model for 120K steps, unfreezing the encoder after 10K steps. Inference is performed with beam size of 5, and the samples with empty ground-truth transcriptions are removed from the evaluation set.

## B. Comparison with Previous Works

Empirical comparisons of MoHAVE with AVMoE (Cheng et al., 2024) and EVA (Wu et al., 2024) are unfortunately infeasible due to fundamental differences in target tasks and methods. Both AVMoE and EVA primarily address audio captioning for visual contexts (*e.g.,* narrating sports game scenes), while our work specifically targets typical AVSR tasks, where both audio and visual inputs directly involve human speech.

Moreover, AVMoE (Cheng et al., 2024) employs a dense MoE; unlike sparse expert structures commonly used in modern LLMs or Transformers, AVMoE's *MoE* is actually implemented as weighting between unimodal and cross-modal adapters, rather than selecting sparse FFN experts. Specifically, AVMoE uses two entirely separate MoEs for audio encoder and visual encoder, infeasible for processing multimodal tokens. Our approach fundamentally differs by employing a sparse multimodal MoE, dynamically routing tokens based on audio-visual inputs.

Closer to our work is EVA (Wu et al., 2024), which simply applies a sparse MoE structure into an audio-visual encoder. Although exact implementation details are unavailable (code and checkpoints unreleased), EVA's structure aligns closely with our basic MoE implementation which we evaluated as *AV-MoE* in Table 2, except ours is in the decoder. As demonstrated in our study (Table 10), applying MoE at the encoder-level like EVA falls behind our multimodal decoder approach. Thus, EVA likely cannot achieve comparable robustness or efficiency.

## C. Expert Group Utilization

### C.1. Group Load Analysis for Entire Layers

In the main paper (Figure 5), we have presented expert load distribution for selected layers. Figure 6 provides the distribution across all MoE layers, illustrating how MoHAVE dynamically adjusts expert groups based on noise conditions.

### C.2. Language-wise Analysis on Multilingual Tasks

We additionally provide language-wise analysis on multilingual AVSR in Figure 7. Our analysis indicates language-dependent differences in the expert allocation within MoHAVE. For example, Arabic tokens tend to be routed more frequently toward

visual experts, whereas Spanish or French tokens rely more heavily on audio experts. However, we also note that these trends vary by layer. Also, within each expert group, the intra-modal router's load-balancing ensures a uniform expert utilization across data samples. Thus, there is no explicit language-specific expert selection within groups, consistent with observations found in Zoph et al. (2022). We suppose that more detailed investigation into expert load distribution across languages and its relation to linguistic/paralinguistic characteristics would serve as valuable future work.

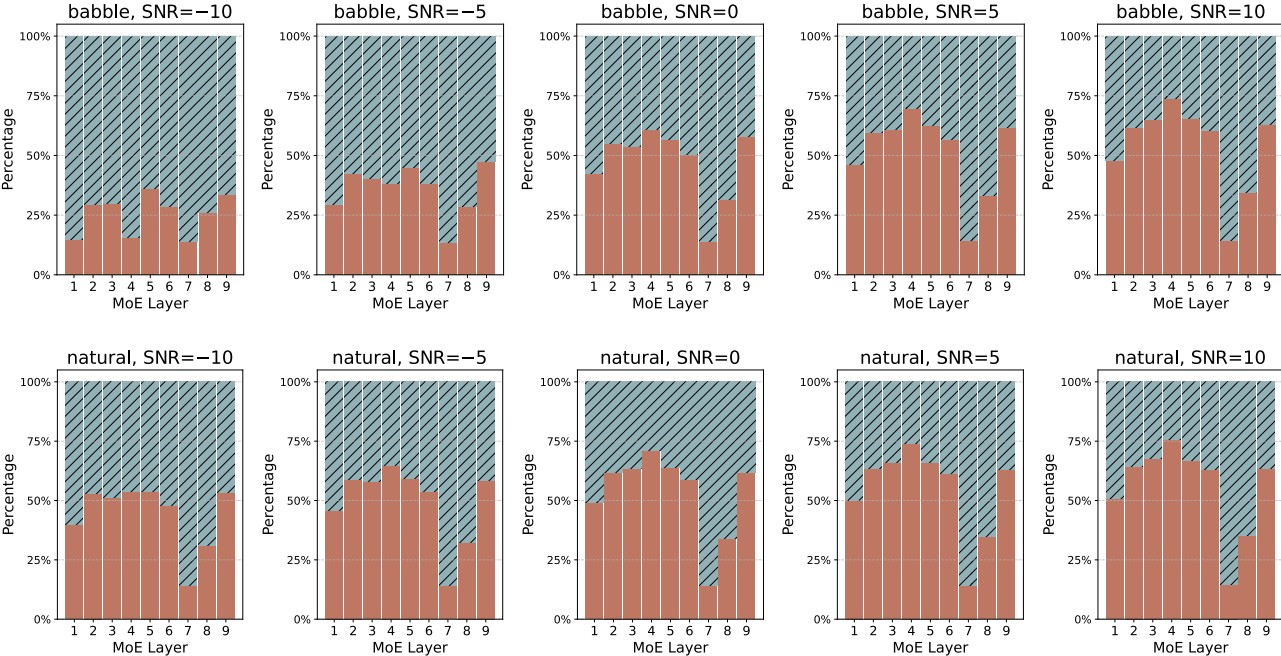

Figure 6. Expert load distribution in MoHAVE for the audio group (solid bars) and visual group (dashed bars) across noisy audio-visual sequences under babble (first row) and natural (second row) noise. The frequency of each expert has been weighted by the inter-modal router's output probability.

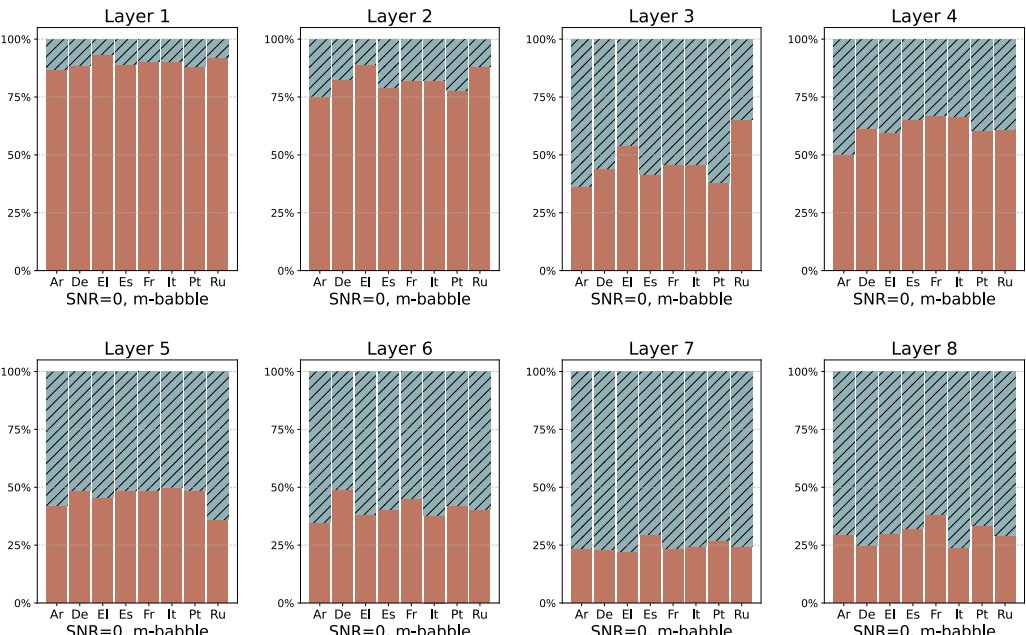

Figure 7. Expert load distribution in multilingual AVSR MoHAVE for the audio group (solid bars) and visual group (dashed bars).

# D. Additional Results

## D.1. Evaluation Under Real-world Noise Conditions

Following the standard practice in robust AVSR works (Hong et al., 2023; Kim et al., 2024), we have introduced various noise conditions (babble, speech, music, and natural) to evaluate our MoHAVE's robustness and adaptability. To better reflect real-world noise conditions, we conduct further evaluations by augmenting LRS3 with realistic background audio from the DEMAND dataset (Thiemann et al., 2013), which contains recordings from diverse indoor and outdoor environments, *e.g.,* cafeteria or meeting room. As summarized in Table 6, on this enhanced benchmark, MoHAVE consistently outperforms AV-HuBERT across various real-world settings, achieving an average WER of 3.6% for 18 environments. These results further confirm MoHAVE's performance under realistic audio-visual conditions.

*Table 6.* Performance comparison on LRS3 (Afouras et al., 2018b) with audio noise sampled from the DEMAND dataset (Thiemann et al., 2013). For each noisy environment, WER (%) is measured by randomly sampling the SNR value from the range $[-10, 0]$ dB.

| Model | PARK | RIVER | CAFE | RESTO | CAFETER | METRO | STATION | MEETING | KITCHEN | LIVING | WASH | FIELD | HALL | OFFICE | SQUARE | TRAFFIC | BUS | CAR | AVG |
|---|---|---|---|---|---|---|---|---|---|---|---|---|---|---|---|---|---|---|---|
| AV-HuBERT-LARGE | 3.8 | 6.1 | 6.4 | 13.1 | 8.6 | 3.1 | 4.7 | 5.7 | **2.2** | 3.6 | **1.6** | **1.7** | 2.1 | 1.9 | 2.7 | 3.0 | 1.9 | **1.6** | 4.1 |
| MoHAVE-LARGE | **3.4** | **4.4** | **5.4** | **11.9** | **6.4** | **3.0** | **4.1** | **4.5** | **2.2** | **3.3** | 1.8 | 1.9 | **1.9** | **1.7** | **2.3** | **2.5** | **1.8** | **1.6** | **3.6** |

## D.2. Multilingual Tasks in Clean Environments

Table 7 outlines the MuAViC benchmark results in a clean environment, without auditory noise added. The experimental setup remains consistent with Table 4, utilizing the same models. Unlike the noisy setting, we observe that MoHAVE does not yield significant performance improvements in clean speech tasks. This is primarily because MoHAVE enhances AVSR under noisy conditions by dynamically adjusting the utilization of audio and visual expert groups.

Indeed, in clean speech recognition and translation tasks, encoder capacity—particularly when pretrained on large-scale audio data—plays a more crucial role than decoder-specific training methods. In addition, visual information is less essential in noise-free environments, as demonstrated by the strong ASR performance of the Whisper-large-v2 model (Radford et al., 2023). Even the smaller ASR model, XLS-R 300M (Babu et al., 2022), surpasses AVSR models such as mAV-HuBERT (Anwar et al., 2023) or u-HuBERT (Hsu & Shi, 2022) in this setting, underscoring that the advantage of using AVSR models emerges most clearly in robust speech recognition.

## D.3. Number of Activated Experts

By default, MoHAVE selects one expert from each group—audio and visual—activating a total of two experts per token. This design is to match the compute of standard MoE implementations, which utilizes top-2 out of 8 experts. A natural question arises: *does activating more experts improve performance, or does it simply increase computational costs without substantial gains?*

*Table 8.* Impact of the number of activated experts on AVSR performance.

| $(k^A, k^V)$ | babble | speech | music | natural | N-WER | C-WER |
|---|---|---|---|---|---|---|
| $(1, 1)$ | 9.6 | 4.2 | 4.7 | 4.5 | 5.8 | 1.8 |
| $(1, 2)$ | 9.3 | 4.1 | 4.8 | 4.4 | 5.7 | 1.9 |
| $(2, 1)$ | 10.1 | 4.5 | 5.3 | 4.9 | 6.2 | 2.4 |
| $(2, 2)$ | 11.0 | 5.2 | 5.8 | 5.5 | 6.9 | 3.0 |

Table 8 presents the results when activating more experts of MoHAVE-BASE, where top-$k^A$ experts from the audio group and top-$k^V$ experts from the visual group are selected. Interestingly, increasing the number of audio experts significantly degrades performance, implying that the model might be confused by employing another sub-optimal expert.

In contrast, activating two visual experts while keeping one audio expert improves performance (N-WER of 5.7%) compared to the default setting of single visual expert. Particularly under the babble noise, WER has decreased from 9.6% to 9.3%. This suggests that adding an additional visual expert can be beneficial in noisy environments, likely due to the increased robustness from visual information in challenging audio conditions.

*Table 7.* Multilingual audio-visual speech task performance, with non-English speech recognition (WER) and X-En speech-to-text translation (BLEU score), in a clean environment without auditory noise. [†]Results obtained from Han et al. (2024). [‡]Re-implemented using the pretrained model from Choi et al. (2024).

| Model | Pretrain data | Ar | De | El | Es | Fr | It | Pt | Ru | Avg |
|---|---|---|---|---|---|---|---|---|---|---|
| | | | | | | Source | | | | |
| *Clean Speech Recognition, Test WER ↓* | | | | | | | | | | |
| Whisper large-v2 (Radford et al., 2023) | *680k hrs, 100+ langs* | 91.5 | **24.8** | 25.4 | 12.0 | 12.7 | 13.0 | 15.5 | 31.1 | 28.2 |
| u-HuBERT[†] (Hsu & Shi, 2022) | *1.7k hrs, English* | 89.3 | 52.1 | 46.4 | 17.3 | 20.5 | 21.2 | 21.9 | 44.4 | 39.1 |
| mAV-HuBERT (Anwar et al., 2023) | *1.7k hrs, English* | 69.3 | 47.2 | 41.2 | 16.2 | 19.0 | 19.8 | 19.9 | 38.0 | 33.8 |
| XLS-R 300M[†] (Babu et al., 2022) | *1.2k hrs, 9 langs* | 85.6 | 44.0 | 34.4 | 13.2 | 15.1 | 14.3 | 16.2 | 34.4 | 32.2 |
| XLAVS-R 300M (Han et al., 2024) | *8.3k hrs, 100+ langs* | 80.0 | 38.0 | 28.1 | 11.7 | 15.3 | 13.8 | 14.4 | 31.2 | 29.1 |
| XLAVS-R 2B (Han et al., 2024) | *1.2k hrs, 9 langs* | 79.3 | 44.4 | **19.0** | **9.1** | **12.3** | **10.6** | **11.2** | **25.0** | **26.4** |
| mAV-HuBERT[‡] | *7.0k hrs, 100+ langs* | **78.3** | 41.4 | 25.5 | 11.9 | 16.2 | 14.8 | 14.3 | 31.6 | 29.3 |
| MoHAVE-LARGE (**ours**) | *7.0k hrs, 100+ langs* | 85.1 | 38.9 | 25.9 | 11.2 | 14.6 | 14.0 | 13.8 | 30.0 | 29.2 |
| *Clean X-En Speech-to-Text Translation, Test BLEU ↑* | | | | | | | | | | |
| Whisper large-v2 (Radford et al., 2023) | *680k hrs, 100+ langs* | - | - | 24.2 | 28.9 | 34.5 | 29.2 | 32.6 | **16.1** | 29.9 |
| mAV-HuBERT (Anwar et al., 2023) | *1.7k hrs, English* | - | - | 7.6 | 20.5 | 25.2 | 20.0 | 24.0 | 8.1 | 17.6 |
| XLAVS-R 300M (Han et al., 2024) | *8.3k hrs, 100+ langs* | - | - | 18.3 | 23.9 | 29.8 | 25.1 | 28.9 | 12.1 | 23.0 |
| XLAVS-R 2B (Han et al., 2024) | *1.2k hrs, 9 langs* | - | - | 21.6 | 25.1 | 30.6 | 26.6 | 29.9 | 13.9 | 24.6 |
| mAV-HuBERT[‡] | *7.0k hrs, 100+ langs* | - | - | 11.5 | 24.2 | 29.2 | 23.9 | 28.1 | 10.4 | 21.2 |
| MoHAVE-LARGE (**ours**) | *7.0k hrs, 100+ langs* | - | - | 13.8 | 24.9 | 30.8 | 25.0 | 28.7 | 10.9 | 22.4 |

## D.4. Unimodal Task Results

Table 9 presents unimodal task results, evaluating model performance on video-only (VSR) sequences and audio-only (ASR) sequences. BRAVEn (Haliassos et al., 2024) and Llama-3.1-8B-AVSR (Cappellazzo et al., 2024) models achieve the best VSR performance, as these models are specifically pretrained for the VSR task. While using an LLM decoder is highly effective in VSR, since LLMs are able to refine and correct recognition errors, ASR performance is largely determined by the encoder's pretraining strategy as BRAVEn and Whisper encoders. As an adaptive audio-visual model, MoHAVE does not specialize in unimodal tasks but instead performs robustly in multimodal AVSR. It only exhibits a slight improvement in VSR over AV-HuBERT. These results indicate that unimodal performance is primarily influenced by the effectiveness of the encoder pretraining strategy rather than the MoE-based multimodal approach.

*Table 9.* Comparison on the unimodal ASR and VSR task performance.

| Method | Encoder + Decoder | V | A |
|---|---|---|---|
| BRAVEn (Haliassos et al., 2024) | BRAVEn + Transformer | 26.6 | 1.2 |
| Llama3.1-8B-AVSR (Cappellazzo et al., 2024) | AV-HuBERT + LLM | 25.3 | 1.4 |
| Llama3.1-8B-AVSR (Cappellazzo et al., 2024) | Whisper + LLM | - | 1.1 |
| AV-HuBERT-LARGE (Shi et al., 2022a) | AV-HuBERT + Transformer | 28.6 | 1.4 |
| MoHAVE-LARGE (**ours**) | AV-HuBERT + MoE | 28.2 | 1.4 |

## D.5. Variations of MoHAVE Implementations

**MoHAVE in the encoder.** We have implemented MoHAVE by integrating MoE into the decoder to facilitate text token processing while enhancing multimodal fusion. Since the AVSR decoder incorporates information from both audio and visual modalities along with text tokens, the decoder-based MoHAVE is expected to be the most effective strategy. An alternative approach is to apply MoHAVE within the encoder, by pretraining the encoder using the AV-HuBERT masked prediction strategy (Shi et al., 2022a). For this, we initialize the pretrained encoder (with standard Transformers) and convert the FFN layers into MoE layers by copying the FFN parameters into all the expert modules. Since the BASE model consists of 12 encoder layers, we convert 6 of them alternatively to match the number of MoE layers in the decoder MoHAVE. During fine-tuning, all MoE layers in the encoder are also trained following the same procedure.

There are two options for pretraining strategies: (1) pretraining only the MoE layers initialized from the FFN parameters,

and (2) pretraining the entire encoder with MoE layers. As shown in Table 10, the latter approach significantly outperforms the former, suggesting that encoder MoHAVE requires full pretraining for effective learning. However, even with full pretraining, encoder MoHAVE performs inferior to decoder MoHAVE. This is because the encoder only processes audio and visual tokens, whereas the decoder directly integrates audio-visual embeddings with text, finding optimal strategies for text token dispatching that best improves speech recognition. In addition, applying MoHAVE to both the encoder and decoder leads to degraded performance despite the increased computational cost.

**Decoder uptraining.** We also explore a successive training strategy for the decoder, referred to as uptraining (Ainslie et al., 2023), where the decoder MoHAVE undergoes additional training after the fine-tuning phase of standard Transformers. However, as seen in Table 10, uptraining does not yield further improvements compared to training from scratch, even after an additional 120K training steps. In fact, we observed the shorter uptraining steps leading to degraded performance. This suggests that training the decoder MoHAVE requires a comprehensive learning phase rather than incremental fine-tuning, as MoE may fundamentally alter the processing pathways of tokens.

*Table 10.* Performance comparison of MoHAVE applied to the encoder, decoder, and both.

| Method | babble | speech | music | natural | N-WER | C-WER |
|---|---|---|---|---|---|---|
| Encoder MoHAVE (pretrain only MoE) | 10.5 | 5.0 | 5.6 | 5.0 | 6.5 | 2.4 |
| Encoder MoHAVE | 10.0 | 4.4 | 5.1 | 4.5 | 6.0 | 1.9 |
| Encoder + Decoder MoHAVE | 10.1 | 4.7 | 5.3 | 4.7 | 6.2 | 2.0 |
| Decoder MoHAVE | 9.6 | 4.2 | 4.7 | 4.5 | 5.8 | 1.8 |
| Decoder MoHAVE (uptrain) | 9.7 | 4.2 | 4.8 | 4.5 | 5.8 | 1.8 |

