# OpenReview forum: "MoHAVE: Mixture of Hierarchical Audio-Visual Experts for Robust Speech Recognition"
_ICML.cc/2025/Conference — ICML 2025 poster_

### Official Review · Reviewer_VMgS · 2025-03-10

**Overall Recommendation:** 3

**Summary:**

MoHAVE is a novel audio-visual speech recognition system that addresses the scalability challenges in traditional AVSR models. The paper introduces a sparse Mixture-of-Experts (MoE) framework combined with a hierarchical gating mechanism that dynamically routes audio-visual inputs to modality-specific expert groups. Here’s a brief summary:

MoHAVE leverages a sparse MoE architecture to scale model capacity while keeping computational costs low. Its hierarchical gating system consists of an inter-modal router that assigns weights to audio and visual expert groups based on input characteristics, and intra-modal routers that further select the top experts within each group. This design enables the model to adapt to varying noise conditions by shifting reliance between modalities—using more visual cues in high auditory noise and vice versa. The paper demonstrates that MoHAVE achieves state-of-the-art performance on robust AVSR benchmarks like LRS3 and in multilingual speech recognition and translation tasks, all while activating only a fraction of its total parameters during inference.

**Claims And Evidence:**

Yes

**Essential References Not Discussed:**

The key contribution is a MOE-based AVSR model, and cite mainstream prior works.

**Experimental Designs Or Analyses:**

Yes

**Methods And Evaluation Criteria:**

Yes

**Other Comments Or Suggestions:**

Na

**Other Strengths And Weaknesses:**

Overall, I think this is a good piece of work; the MoE technique is well-suited for AVSR, and previous studies have shown that simply fusing the two modalities can lead to models that overly depend on audio. From an AVSR perspective, the experimental results achieved in this paper are excellent, representing a robust algorithm. However, since my own research field does not focus on MoE, I cannot provide a particularly meaningful evaluation of the innovations in the MoE component.

**Questions For Authors:**

Na

**Relation To Broader Scientific Literature:**

A scalable sparse MoE framework tailored for audio-visual speech recognition.
A novel hierarchical gating mechanism that dynamically adjusts expert group utilization based on input context.
Superior robustness and accuracy under noisy conditions, setting new performance benchmarks on standard AVSR tasks.

**Theoretical Claims:**

Yes

---

> ### Author Rebuttal · Authors · 2025-03-29
>
> Thank you for reviewing our paper thoroughly and for recognizing the key strengths, including the **suitable hierarchical MoE architecture for AVSR, a novel gating mechanism, and strong experimental results demonstrating robust performances**.
>
> We greatly appreciate your thorough evaluation from the perspective of AVSR, even though MoE lies outside your primary research field. As such, we would like to briefly highlight our contributions of MoHAVE to the MoE framework:
>
> - **Novel hierarchical gating mechanism**: An inter-modal router dynamically allocates tokens to audio or visual expert groups based on input characteristics, and an intra-modal router further dispatches tokens to appropriate experts within these groups.
> - **Adaptive and robust MoE**: Previous works for multimodal MoEs have relied on modality fusion strategies or assigning fixed roles to each expert. While they lack adaptability and robustness to dynamically changing noisy environments, our work addresses this limitation by dynamically adjusting the usage of expert groups.
>
> We also kindly encourage you to go through other reviewers' comments as well as our responses, and if any additional concerns arise, please let us know through the discussion phase. We would be glad to provide any further clarification.

---

### Official Review · Reviewer_Fwbw · 2025-03-13

**Overall Recommendation:** 4

**Summary:**

This paper proposes an adaptive hierarchical routing mechanism in mixture of experts model for audio-visual (AV) speech recognition and AV language X speech to English text translation. As compared to a hard routing of modalities into modality specific expert groups, this paper uses a combination of inter-modal and intra-modal routers while still keeping audio and video experts separate. This way the model can utilize A-only, V-only and AV data in one model. The paper also implements some load balancing techniques to balance the mixture weights.

For AVSR, LRS3 dataset is used and the model performance is reported under various noise conditions. For cross-lingual speech task, MuAViC benchmark is used. The model is compared agains Base AV MoE method and AV-HuBERT method. In the AVSR experiments, it is shown that the WER on noisy datasets improve over AV-HuBERT and that the load biasing is a crucial implementation detail that affect the final WER. In the X-En Speech-to-Text Translation, the large MoHAVE model outperforms XLAVS-R models for most of the test languages in terms of the BLEU score.

## update after rebuttal
I would like to keep my score after rebuttal.

**Claims And Evidence:**

Yes

**Essential References Not Discussed:**

NA

**Experimental Designs Or Analyses:**

Went through the tables but have not checked the details of Figs. 4 and 5 (which have an expected pattern of relying less on audio and more on video in low SNR conditions).

One of the main contributions is presented as "robust AVSR benchmarks including multilingual tasks, delivering high accuracy while maintaining computational overhead. " However, the computational efficiency is only discussed in the appendix. It could have been useful to include that analysis in the main text and Fig. 4 or 5. could have been moved to the Appendix.

**Methods And Evaluation Criteria:**

Yes

**Other Comments Or Suggestions:**

1) Eq. 11, please check the notation as described above

2) One of the main contributions is presented as "robust AVSR benchmarks including multilingual tasks, delivering high accuracy while maintaining computational overhead. " However, the computational efficiency is only discussed in the appendix. It could have been useful to include that analysis in the main text and to save space Fig. 4 or 5. could have been moved to the Appendix. This is a minor suggestion.

**Other Strengths And Weaknesses:**

+ Strengths: A variant of the MoE routing is proposed which might be useful for other multimodal applications. Results show some improvement over the AV-HuBERT baseline for AVSR and MoHAVE also can handle some multilingual tasks.

- Weaknesses: The notation in the equations could be impoved, and the paper might benefit from including the computational cost analysis in the main text.

**Questions For Authors:**

- What was the reason for the following choice? "For sequences containing both audio and video, we exclude them from the load biasing loss calculation but incorporate them into the load balancing."

- How are the batches constructed during training? Do the batches contain a mix of audio-only datapoints, video-only datapoints and AV data points in a single batch?

**Relation To Broader Scientific Literature:**

In line with previous studies, load balancing is crucial for the success of the MoE implementation. The analysis of the mixture weights per modality was also supporting earlier observations around relying more on the visual component when the acoustic noise is heavy.

Since multimodal systems are becoming more popular, the proposed hierarchical multimodal MoE might be helpful for other studies in other multimodal applications.

**Theoretical Claims:**

Equations in the main text mostly seem to be correct. In Eq. (11), $E^{A}$ is used both as the set of experts and the number of experts in summation, hence the notation is a little confusing.

---

> ### Author Rebuttal · Authors · 2025-03-29
>
> > Weakness 1: The notation in the equations could be improved
>
> A1: Thank you for the detailed review. We will revise the notations, especially in Eq. (11), to clearly distinguish the set of experts from its cardinality.
>
>
> > Suggestion 1: The computational cost analysis is only discussed in Appendix
>
> A2: We agree with your suggestion that computational efficiency is indeed one of the main contributions of MoHAVE, especially given its sparse MoE architecture that activates only a fraction of parameters. To better highlight this contribution, we will move the computational cost comparison as well as its discussion currently presented in Appx. A.2 into the main body.
>
> > Question 1: What was the reason for the following choice? *For sequences containing both audio and video, we exclude them from the load biasing loss calculation but incorporate them into the load balancing.*
>
> A3: The load biasing loss is designed specifically to encourage modality specialization for a subset of experts. Audio-visual multimodal tokens do not inherently favor one modality-specific expert group over another. Therefore, these tokens are included in the load balancing, ensuring uniform expert group loads across every multimodal token.
>
> > Question 2: Do the batches contain a mix of audio-only, video-only, and AV data points in a single batch?
>
> A4: Yes, each batch is constructed by sampling a total of 320 seconds of data, structured as follows: 25% of the data points in the batch are audio-only (video dropped), another 25% are video-only (audio dropped), and the remaining 50% are multimodal, containing both audio and video.

---

### Official Review · Reviewer_QYg3 · 2025-03-13

**Overall Recommendation:** 3

**Summary:**

The paper introduces MoHAVE, a novel Audio-Visual Speech Recognition (AVSR) framework leveraging a Mixture of Experts (MoE) architecture. By dynamically selecting modality-specific experts through a hierarchical gating mechanism. Experimental results on benchmark datasets demonstrate its effectiveness, outperforming existing models in challenging noisy environments.

**Claims And Evidence:**

The main claims of the paper are:

1. MoE architecture for scaling AVSR systems that effectively scales AVSR models while maintaining computational efficiency
2. Hierarchical gating for adaptive expert utilization
3. Robust AVSR performance

These claims are generally well-supported by experimental results. However, a comparison with all relevant baseline models (e.g. CMA, UniVPM) is important to support claim 3.

**Essential References Not Discussed:**

NA

**Experimental Designs Or Analyses:**

The experimental design and analysis are well-structured with the use of two AVSR benchmarks: LRS3 and MuAViC. The authors compare their approach with multiple baseline models (e.g. AV-HuBERT, AV-MoE),  perform ablation analysis (load biasing loss, hard routing, and number of activated experts), along with a computational cost analysis that show the efficiency of the proposed model. However, some baseline models (e.g. CMA, UniVPM) are not included in Table 1 for comparison, despite being mentioned in the related work and included in Table 2.

**Methods And Evaluation Criteria:**

The proposed method and evaluation criteria are well-defined. The hierarchical Mixture-of-Experts (MoE) approach is a reasonable architectural choice for improving model scalability and the evaluation benchmarks (LRS3 and MuAViC) are widely used in the AVSR research community. However, the evaluation mostly relies on synthetic noise additions rather than real-world conditions. It would be interesting to show how the model performs on real-world data.

**Other Comments Or Suggestions:**

NA

**Other Strengths And Weaknesses:**

Strengths

1. Scalability without excessive computational cost.
2. Adaptive expert selection improves generalization across different noise conditions.
3. Comprehensive benchmarking across various AVSR datasets and multilingual settings.

Weaknesses

1. The model has been evaluated on synthetic data and not on real-world conditions
2. Some relevant baseline models (e.g. CMA, UniVPM) are missing from the comparison.

**Questions For Authors:**

1. Why don’t you compare MoHAVE with the CMA (Kim et al., 2024) and UniVPM (Hu et al., 2023c)  in Table 1?
2. Can MoHAVE handle asynchronous speech and lip movements (e.g. delays in video frames relative to audio)?
3. How do different languages affect expert selection and the hierarchical MoE routing?

**Relation To Broader Scientific Literature:**

The paper’s key contributions build upon prior work in Audio-Visual Speech Recognition (AVSR) using Mixture-of-Experts (MoE), including models like AV-HuBERT, AV-data2vec, and Auto-AVSR, which utilize self-supervised learning for audio-visual speech processing. However, MoHAVE advances this approach by incorporating a hierarchical MoE framework, enhancing scalability and robustness while maintaining computational efficiency.

**Theoretical Claims:**

The theoretical claims are well-supported particularly in the discussion of MoE routing mechanisms and hierarchical gating. The paper clearly defines the load balancing loss and load biasing loss to improve expert selection. The hierarchical gating strategy for inter-modal and intra-modal routing is mathematically justified, and the empirical results show effectiveness the approach.

---

> ### Author Rebuttal · Authors · 2025-03-29
>
> > Weakness 1: The model is evaluated on synthetic data and not on real-world conditions
>
> A1: We acknowledge your concern regarding evaluation with synthetic noise data. While standard AVSR benchmarks such as LRS3 and MuAViC typically offer curated datasets with high-quality audio and clear visual information, **these benchmarks alone cannot fully represent real-world noisy conditions.** Therefore, following standard practice in robust AVSR research [1,2], we have introduced various noise conditions to evaluate our MoHAVE's robustness and adaptability.
>
> Additionally, to better reflect real-world noise conditions, we conducted further evaluations by augmenting LRS3 with realistic background audio from the DEMAND dataset [3], which contains recordings from diverse indoor and outdoor environments, e.g., cafeteria. On this enhanced benchmark (at SNR=-10~0), MoHAVE consistently outperformed AV-HuBERT across various real-world settings including **cafeteria (WER: 6.4 vs. 8.6), restaurant (11.9 vs. 13.1), meeting room (4.5 vs. 5.7), and river (4.4 vs. 6.1)**, achieving an **average WER of 3.6% vs. 4.1%** across all 18 environments. These results further confirm MoHAVE’s performance under realistic audio-visual conditions.
>
>
> > Weakness 2: CMA and UniVPM are missing from Table 1
>
> A2: The CMA and UniVPM models in Table 2 are all built upon AV-HuBERT-Large, **matching the architecture and activated parameter count with the dense (non-MoE) baseline in Table 1.** Furthermore, both Table 1 and Table 2 evaluate under identical experimental setups and noise configurations (i.e., babble, speech, music, and natural noises), while Table 2 reports average results for music and natural. For detail, please refer to the table below.
>
> Method|# Experts|Groups|Activated Params|Total Params|babble|speech|music|natural|N-WER|C-WER
> -|-|-|-|-|-|-|-|-|-|-
> UniVPM|-|-|478M|478M|9.3|4.1|3.6|3.6|5.2|1.2
> CMA|-|-|500M|500M|8.1|2.9|3.8|3.6|4.6|1.5
>
> We note that Table 1 primarily demonstrates the effectiveness and efficiency of MoHAVE, **by comparing different MoE variants**, including standard MoE, hard routing, and hierarchical MoE. In contrast, CMA and UniVPM in Table 2 utilize special modules for cross-modality, which are orthogonal to the MoE framework, focusing instead on audio-visual fusion or feature enhancement strategies independent of expert routing mechanisms. Yet, recognizing the importance of comprehensive comparisons, we have included the result of incorporated CMA + MoHAVE in Table 2.  To improve clarity, we will revise by merging Table 1 and Table 2.
>
> > Question 1: Can MoHAVE handle asynchronous speech and lip movements?
>
> A3: We have not yet evaluated MoHAVE under audio-visual asynchronous conditions. Our current framework is optimized for scenarios where audio and video are temporally aligned, as is standard in most AVSR works. Prior works that address the audio-visual asynchrony have proposed solutions like external synchronization module [4], which explicitly model temporal offsets between audio and visual streams. While MoHAVE does not currently model asynchrony, we believe combining MoHAVE with methods that explicitly handle asynchronous inputs could be a valuable extension for future research.
>
>
> > Question 2: How do different languages affect expert selection and hierarchical MoE routing?
>
> A4: Thank you for this insightful question. Our analysis indicates language-dependent differences in expert allocation within MoHAVE. For example, Arabic tokens tend to be routed more frequently toward visual experts, whereas French or Spanish tokens rely more heavily on audio experts (please see this [**anonymized link**](https://figshare.com/s/a45cbf35654ae37b2ce9)). However, we also note that these trends vary by layer. Also, within each expert group, the intra-modal router’s load-balancing ensures a uniform expert utilization across data samples. Thus, there is no explicit language-specific expert selection within groups, consistent with observations found in [5]. We suppose that more detailed investigation into expert load distribution across languages and its relation to linguistic/paralinguistic characteristics would be valuable future work.
>
> ---
> References:
>
>
> [1] Hong et al. "Watch or listen: Robust audio-visual speech recognition with visual corruption modeling and reliability scoring." CVPR, 2023.
>
>
> [2] Kim et al. "Learning video temporal dynamics with cross-modal attention for robust audio-visual speech recognition." SLT, 2024.
>
>
> [3] Thiemann et al. "The diverse environments multi-channel acoustic noise database" Proceedings of Meetings on Acoustics, 2013.
>
>
> [4] Li et al. "Unified cross-modal attention: robust audio-visual speech recognition and beyond." TASLP, 2024.
>
>
> [5] Zoph et al. "St-moe: Designing stable and transferable sparse expert models." arXiv, 2022.

---

### Official Review · Reviewer_ikv4 · 2025-03-17

**Overall Recommendation:** 2

**Summary:**

This paper enhances audio-visual speech recognition based on MoE with audio/visual hierarchical modeling. This paper attaches the audio-visual MoE fusion parts to the decoder part and controls the audio and video expert groups, respectively, based on the group-level load biasing loss so that each modality contributes to audio-visual speech recognition in a balanced manner. The experimental results show that this balanced (hierarchical) manner performs reasonably from the hard routing-based approaches. The method also achieved state-of-the-art performance in the noisy LRS3 benchmark and multilingual audio-visual speech recognition/translation.

## update after rebuttal
I checked the rebuttal, but I still could not fully convince myself that this paper has significant novelty from AV-MoE (Cheng et al.) and EVA (Wu et al.) in terms of the methodology, as the major concept is similar. The actual experimental comparisons will make their claims stronger. Thus, I want to keep my score as it is.

**Claims And Evidence:**

- The proposed MoHAVE (Mixture of Hierarchical Audio-Visual Experts) shows performance improvements from hard routing-based approaches. However, its margin is small.
- The method shows the state-of-the-art performance on two public benchmarks (audio-visual noisy speech recognition based on LRS3 and multilingual audio-visual speech recognition/translation based on the MuAViC benchmark)
- The paper also shows how much each modality contributes by checking the MoE posterior values, but this does not have comparisons, and it is difficult to validate better interpretability.

**Essential References Not Discussed:**

The paper references prior AV speech recognition methods based on MoE (Cheng et al., 2024; Wu et al., 2024) and distinguishes its focus on general and speech video scenarios. However, this distinction is insufficient, as all approaches (MoHAVE, Cheng et al., 2024; Wu et al., 2024) are applicable to both scenarios. Given the methodological similarities, the paper should include experimental comparisons with these prior works to strengthen its claims.

**Experimental Designs Or Analyses:**

- I found that the straightforward approaches based on the hard-routing method are very competitive. The paper should provide the hard-routing method in the following experiments (e.g., Tables 2 and 3).
- The expert load distribution results (Figures 4 and 5) are intuitive and interpretable. However, there are no comparisons, and it is difficult to discuss whether this behavior is better or not. Is it possible to compute that of the hard-routing method? Then, we can discuss how the proposed MoHAVE is more reasonable.

**Methods And Evaluation Criteria:**

- The paper uses two public benchmarks (audio-visual noisy speech recognition based on LRS3 and multilingual audio-visual speech recognition/translation based on the MuAViC benchmark). It is based on the established token (word/character) error rate and BLEU score, and its evaluation criteria are valid.

**Other Comments Or Suggestions:**

- Equation (2): $E_i(x)$ suddenly appears without any explanation. Also, it is confusing because $E$ is used as the number of experts in equation (1). I recommend the authors rewrite these equations.
- This is just a note. The TED organization changed its policy last year and did not want researchers to use the TED data for AI development. The MuAViC dataset is based on the TED data, and it would be difficult to publish the results based on this data in the future.

**Other Strengths And Weaknesses:**

Strengths
- showing the state-of-the-art performance in two public audio-visual speech recognition/translation benchmarks, including the multilingual setup
- the modality contribution is intuitive

Weaknesses
- the proposed method and its improvement are incremental.
- hard-routing methods seem to be strong, and the paper should have more comparisons with them
- AV ASR has already been studied in various literature (e.g., (Cheng et al., 2024; Wu et al., 2024)) and its contribution is marginal. The experimental comparisons with (Cheng et al., 2024; Wu et al., 2024) may somewhat mitigate this weakness.

**Questions For Authors:**

Figures 4 and 5: Can you compute and compare the expert load distribution in the hard-routing methods versus the proposed methods? If the proposed methods demonstrate greater interpretability, this would further substantiate the method’s interpretability claims.

**Relation To Broader Scientific Literature:**

- multi-modal processing (vision, text, and speech) has become very important in recent AI technologies. Also, MoE has become a very active research topic in ML now. So, this research that combines them would have a good broader scientific impact.

**Theoretical Claims:**

- This paper does not have a theoretical component.

---

> ### Author Rebuttal · Authors · 2025-03-29
>
> > Weakness 1: The proposed method and its improvement are incremental
>
> A1: Thank you for your valuable comments. We would like to clarify that MoHAVE introduces several key innovations in both scalability and robustness for AVSR systems, which go beyond existing works:
>
> MoHAVE is the **first AVSR framework that scales up to ~1B params**, through using a sparse MoE architecture to enable efficient scaling with low computational overhead. To mitigate the model’s inherent bias toward audio, we also introduced **expert group specialization**, followed by a **novel hierarchical gating mechanism** that dynamically routes tokens based on modality reliability and input characteristics. Unlike previous multimodal MoEs relying on modality fusion or assigning fixed roles to each expert, MoHAVE explicitly adjusts expert group usage dynamically—enhancing both adaptability and robustness.
>
> MoHAVE achieves **state-of-the-art results across robust AVSR benchmarks** (Tables 2 and 3). While the improvements in Table 1 may seem modest on average, the gains under severe noise conditions are substantial. As shown in Appx. A.3 (Table 5), MoHAVE-Large achieves 5.0% WER on LRS3 with speech noise at SNR=-10—yielding a **56.1% relative WER improvement over AV-HuBERT-Large, 36.7% over AV-MoE-Large, and 25.4% over the hard-routing variant.** This indicates that MoHAVE correctly predicts over half of the words AV-HuBERT misses.
>
> We believe these contributions pose a significant breakthrough in AVSR scalability and adaptive learning, and our hierarchical routing design offers broad potential for other multimodal MoE applications as well.
>
> > Weakness 2: The paper should have more comparisons with hard routing
>
> A2: We initially introduced hard routing into AVSR to utilize visual experts for noise robustness. However, it lacks adaptability since the allocation of expert groups must be manually fixed (50% audio / 50% visual in our implementation). This approach is sub-optimal depending on audio-visual input quality (as discussed in Fig. 4(b)), and the limitation becomes clearer under challenging environments. Under severe conditions (e.g., Table 5, SNR=-10, -5) hard routing cannot dynamically adjust expert usage, performing much worse than MoHAVE. In response to the reviewer’s suggestion, we additionally evaluated the hard routing model on MuAViC, as **provided below**. Here, hard routing substantially underperforms MoHAVE and even mAV-HuBERT.
>
> Model (Task)|Ar|De|El|Es|Fr|It|Pt|Ru|Avg
> -|-|-|-|-|-|-|-|-|-
> Hard Routing (AVSR)|93.4|49.3|35.7|20.3|23.6|23.4|24.1|44.7|39.3
> **MoHAVE** (AVSR)|92.9|47.3|35.3|18.7|21.2|21.6|21.9|40.6|**37.4**
> Hard Routing (AVS2TT)|-|-|6.7|19.9|24.7|19.6|23.0|7.2|16.8
> **MoHAVE** (AVS2TT)|-|-|11.4|22.3|27.1|22.1|25.1|9.2|**19.5**
>
> **Regarding Fig. 4 and 5:** computing expert load distribution for hard routing would be trivial. By design, expert usage is manually set depending on the input: 100% audio experts for audio-only, 100% visual for video-only, and 50/50 for audio-visual (finding optimal split is heuristic). Unlike MoHAVE, there is no data-driven or noise-aware expert selection. Thus, hard routing would trivially display static distributions without dynamic behavior.
>
> > Weakness 3: Experimental comparisons with (Cheng et al., 2024; Wu et al., 2024)
>
> A3: Direct comparisons with AV-MoE (Cheng et al.) and EVA (Wu et al.) are unfortunately infeasible due to fundamental differences in target tasks and methods. Both AV-MoE and EVA primarily address audio captioning for visual contexts (e.g., narrating sports game scenes), while our work specifically targets typical AVSR tasks, where both audio and visual inputs directly involve human speech.
>
> Moreover, AV-MoE employs a dense MoE; unlike sparse expert structures commonly used in modern LLMs or Transformers, AV-MoE’s "MoE" is actually implemented as weighting between unimodal and cross-modal adapters, rather than selecting sparse FFN experts. Specifically, AV-MoE uses two entirely separate MoEs for audio encoder and visual encoder, infeasible for processing multimodal tokens. Our approach, MoHAVE, fundamentally differs by employing a sparse multimodal MoE, dynamically routing tokens based on audio-visual inputs.
>
> Closer to our work is EVA, which simply applies a sparse MoE structure into an audio-visual encoder. Although exact implementation details are unavailable (code/checkpoints unreleased), EVA’s structure aligns closely with our basic MoE implementation which we evaluated as *AV-MoE* in Table 1 (*AV-MoE-Base* and *AV-MoE-Large*), except ours is in the decoder. As demonstrated in our study (Table 9 in Appx. B.5), applying MoE at the encoder-level—like EVA—falls behind our multimodal decoder approach. Thus, EVA likely cannot achieve comparable robustness or efficiency.
>
> > Suggestion 1: Rewrite the equations
>
> A4: Thank you. We will revise the equations to clearly distinguish the expert set from its cardinality, defining $E_i$ as the output of the $i$-th expert.

---

### Decision · Program_Chairs · 2025-05-01

**Decision:**

Accept (poster)

**Comment:**

The authors propose a MoHAVE (Mixture of Hierarchical Audio-Visual Experts) framework in this paper for audio-visual speech recognition and translation. A sparse MoE architecture is introduced to efficiently scales up the AVSR model capacity. The hierarchical gating design which includes inter-model and intra-model routing can dynamically assign the expert groups for different modalities based on the input context. The authors report the SOTA performance of the proposed MoHAVE on AVSR benchmark LRS3 and multilingual AV speech recognition and translation benchmark MuAViC. Overall, all reviewers consider the work interesting and performance of MoHAVE is good although the technical novelty of using MoE for AVSR is not overwhelmingly significant. The authors' rebuttal has cleared up most of the concerns raised by the reviewers.